# Reconstruction of Compton Edges in Plastic Gamma Spectra Using Deep Autoencoder

**DOI:** 10.3390/s20102895

**Published:** 2020-05-20

**Authors:** Byoungil Jeon, Youhan Lee, Myungkook Moon, Jongyul Kim, Gyuseong Cho

**Affiliations:** 1Intelligent Computing Laboratory, Korea Atomic Energy Research Institute, Daejeon 34507, Korea; bijeon@kaeri.re.kr (B.J.); youhanlee@kaeri.re.kr (Y.L.); 2Department of Nuclear and Quantum Engineering, Korea Advanced Institute of Science and Technology, Daejeon 34141, Korea; 3Quantum Beam Science Division, Korea Atomic Energy Research Institute, Daejeon 34507, Korea; moonmk@kaeri.re.kr (M.M.); kjongyul@kaeri.re.kr (J.K.)

**Keywords:** plastic gamma spectra, energy broadening correction, Compton edge reconstruction, deep learning, deep autoencoder

## Abstract

Plastic scintillation detectors are widely utilized in radiation measurement because of their unique characteristics. However, they are generally used for counting applications because of the energy broadening effect and the absence of a photo peak in their spectra. To overcome their weaknesses, many studies on pseudo spectroscopy have been reported, but most of them have not been able to directly identify the energy of incident gamma rays. In this paper, we propose a method to reconstruct Compton edges in plastic gamma spectra using an artificial neural network for direct pseudo gamma spectroscopy. Spectra simulated using MCNP 6.2 software were used to generate training and validation sets. Our model was trained to reconstruct Compton edges in plastic gamma spectra. In addition, we aimed for our model to be capable of reconstructing Compton edges even for spectra having poor counting statistics by designing a dataset generation procedure. Minimum reconstructible counts for single isotopes were evaluated with metric of mean averaged percentage error as 650 for ^60^Co, 2000 for ^137^Cs, 3050 for ^22^Na, and 3750 for ^133^Ba. The performance of our model was verified using the simulated spectra measured by a PVT detector. Although our model was trained using simulation data only, it successfully reconstructed Compton edges even in measured gamma spectra with poor counting statistics.

## 1. Introduction

Plastic scintillation detectors have poor spectroscopic characteristics because of poor energy resolution and absence of photo peak in the region of interest, which is above 100 keV. Therefore, it is hard to conduct radioisotope identification from plastic gamma spectra. Despite their weaknesses, plastic scintillation detectors have been widely used in radiation monitoring systems, e.g., radiation portal monitor, because they have unique characteristics such as low cost, are easily made in large volume, etc. Therefore, various spectral processing techniques have been reported for pseudo gamma spectroscopy of plastic scintillation detectors. Energy windowing [1,2,3,4], F-score analysis [5], energy weighted algorithms [6,7], and inverse matrix [8] are representative methods for pseudo gamma spectroscopy. However, these methods can be categorized as indirect pseudo gamma spectroscopic methods because it is impossible to directly identify the energy of incident gamma rays. Even though inverse matrix allows unfolding photo peaks in plastic gamma spectra, it works with spectra with good counting statistics only.

In contrast, there have been many studies on radioisotope identification, which is one of the purposes of gamma spectroscopy, using pattern recognition methods, such as library matching [9,10] and neural network-based classifiers [11,12]. Using library matching methods, it is possible to identify radioisotopes only if the library data are prepared to match with the measured data. In the case of neural network-based-classifiers, it is difficult to define practical accuracy. Although the outputs from neural networks are in the form of probabilities, they do not represent practical accuracy without confidence calibration [13].

In this paper, we propose a deep autoencoder model to correct the energy broadening effect, which is one of the weaknesses of the plastic gamma spectra. If the energy broadening effect is corrected, it is possible to conduct direct pseudo gamma spectroscopy differently from other methods because Compton edges are represented in gamma spectra. The datasets for this study were generated using the following procedure; establishment of probabilistic density function (PDF) library for radioisotopes using the results of Monte Carlo simulations, synthesis of PDFs with dependent random ratios for various combinations of radioisotopes, and generation of datasets via random sampling. For the generated and measured plastic gamma spectra, it has been verified that our model can reconstruct Compton edges from spectral measurement, even from spectra with low counting statistics.

## 2. Materials and Methods

### 2.1. Deep Autoencoder

An autoencoder is a type of an artificial neural network that generates an output signal whose dimension is identical to that of the input signal. Figure 1 shows a schematic of autoencoder architecture [14,15,16]. As shown in Figure 1, an autoencoder consists of two parts: encoder and decoder. In the encoder, inputs are encoded into internal representations with reduced dimensions in the latent space. In the decoder, internal representations are decoded into the reconstructed signal. In this unsupervised manner, the autoencoder is widely used for dimension reduction in many applications [17,18]. Furthermore, an autoencoder can be used for noise rejection. If we add noise signals to training data and train an autoencoder to reconstruct the input signal without the noise, the autoencoder is optimized to make a function to reject noise signals. A deep autoencoder is an autoencoder model whose encoder and decoder consist of three hidden layers or more [19].

### 2.2. Establishment of Datasets

#### 2.2.1. Experimental Set-up

EJ-200 (cylindrical shape, dia. 30 × 50 mm, EJ technology) coupled with a PMT (R2228, HAMAMATSU) [20] and a preamp (E990-501, HAMAMATSU) [21] was used as a plastic scintillation detector. Optical grease (BC630, Saint-Gobain, Courbevoie, France) was applied at the junction between the crystal and PMT for optical coupling. For optical shielding, the crystal was wrapped with Teflon and black friction tape. A pulse processor (DP5G, Amptek, Hawthorne, NJ, USA) was used as a shaping amp with time constant of 2.2 μs and multichannel analyzer. A high-voltage supplier (NHQ 224M, ISEG, Lisboa, Portugal) was used to supply operating voltage to the detector. Experiments to measure gamma spectra were conducted in an aluminum dark box for the replenishment of optical shielding. The dark box consisted of a 10 mm thick aluminum case with an internal space of 440 × 440 × 899 (W × H × L) mm. The detector was placed on the shelf of the dark box, and the window of the detector was located at the center of the dark box. ^22^Na, ^60^Co, ^133^Ba, and ^137^Cs were used as gamma ray sources, and the position of the source was fixed at 5 cm from the detector window. Figure 2 shows our experimental setup. Energy calibration was conducted using a parametric optimization method [22].

#### 2.2.2. Monte Carlo Simulation

To simulate plastic gamma spectra, we implemented a simulation geometry that was analogous to the experimental setup using the MCNP 6.2 software [23]. Compositions and densities of materials were defined from a material data report [24]. Gamma ray sources were defined as point sources. An F8 tally was used to simulate the spectral response of each source, and history number was set to 10^8^. The F8 tally is also called a pulse height tally, and it is utilized when simulating deposited energy distribution according to energy bins, time bins, etc. Herein, we use the F8 tally with defining energy bins to simulate spectral response of our plastic scintillation detector. Energy bins for the F8 tally were defined as identical to energy calibrated channel bins. To acquire ideal and energy broadened pulse height distributions, F8 tallies were defined with and without a Gaussian energy broadening (GEB) card to acquire ideal and energy-broadened pulse height distributions, respectively. Coefficients “a”, “b”, and “c” for the GEB card were calculated by a parametric optimization method [22] using experimental spectra that were analogous to the measurement data to the maximum extent. Coefficient used for the GEB card is 0.006779 for “a”, 0.3549 for “b”, and −0.4999 for “c”. 

In MCNP 6.2, the energy broadening effect can be simulated with the use of a GEB option. When the GEB option is activated, all particle histories tallied in F8 tally are recorded after random sampling, which follows Gaussian probability distributions calculated by Equation (1):(1)$$f\left(E_{0},a,b,c\right)=Ae^{-{\left(\frac{2\sqrt{2\ln2}\left(E-E_{0}\right)}{a+b\sqrt{E_{0}+cE_{0}^{2}}}\right)}^{2}}$$
where *A* is a normalization constant; *a*, *b*, and *c* are GEB parameters; *E* is the broadened energy; and *E*_0_ is the original energy before broadening.

#### 2.2.3. Dataset Generation

The datasets were generated by random sampling and data synthesis using simulation data only. Before dataset generation, we prepared libraries of PDFs for ideal and GEB cases as follows. For pulse height spectra of ^22^Na, ^60^Co, ^133^Ba, and ^137^Cs simulated by MCNP code, each spectrum was divided by the integral value of itself for data normalization. With this procedure, each normalized spectrum could be represented as a PDF of detector response, because the summation of each normalized spectrum is one. After PDF libraries were created, we generated datasets as follows. First, ratios for PDF synthesis were selected as significant figures with first decimal place by dependent random sampling; the summation of synthesis ratios should be one to keep the synthesized results as PDFs. Some examples to explain the characteristics of dependent random ratios are as follows. If the synthesis ratio for ^22^Na is one, the ratios for others should be zero. If the ratio for ^22^Na is 0.1, the ratio for ^60^Co is determined in the range of 0 to 0.9. If the ratio for ^60^Co is determined as 0.5, the ratio for ^133^Ba is selected in a range of 0 to 0.5. If the ratio of ^133^Ba is 0, the ratio of ^137^Cs is 0.4. With this spectral synthesis, data for multiple radiation sources in various ratios can be generated without additional simulation. Second, the number of samplings to simulate spectral data was then selected in the range of 40,000 to 100,000. By randomly selecting the sampling numbers, datasets with various levels of statistical uncertainties could be generated. This means that it is possible to build an autoencoder model with the ability to reconstruct Compton edges even from spectra with poor counting statistics with the generated datasets. Once the synthesis ratios and number of samplings were determined, PDFs were synthesized for ideal and GEB cases, and spectra were simulated via random sampling with the synthesized PDFs and the determined number of samplings. Next, spectra were normalized by total sum normalization, which can be represented as Equation (2),
(2)$$x_{i,norm}=\frac{x_{i}}{\sum_{i=1}^{n}x_{i}}$$
where $x_{i}$ is the *i*th element of the original data *X*, $x_{i,norm}$ is *i*th element of the normalized data *X_norm_*, and *n* is the number of elements in the original data set.

In this manner, we established a procedure to generate datasets for the ideal case and GEB case paired with each other. Figure 3 illustrates the dataset generation procedure. With the established dataset generation procedure, we generated 60,000 spectra as a training set, 2000 spectra as a validation set, and 2000 spectra as a test set. Figure 4 shows the examples of the generated datasets.

## 3. Results

### 3.1. Results for Compton Edge Reconstruction with Test Set

The deep autoencoder was implemented in the Python environment using the Tensorflow [25] and KERAS [26] libraries. Hyperparameters for our autoencoder model were determined by trial and error as follows. The architecture of our model consists of three hidden layers as the encoder and three hidden layers as the decoder. The dimension of the input layer is 500, which means spectral data with 500 channel bins are provided as input to the autoencoder. The numbers of neurons in encoder layers are 200, 100, and 50, and the numbers of neurons in the decoder layers are 100, 200, and 500. This means that the input data are compressed by internal representations with dimension of 50 bins during the encoding process, and output with dimension of 500 bins is reconstructed from internal representations during the decoding process. For activation functions of hidden layers, a ReLU function was used for all layers of the encoder and the first and second layers of the decoder. For the third layer of the decoder, a sigmoid function was used as the activation function. 

To train the deep autoencoder, training and validation sets for GEB case were given as input, and those for the ideal case were given as desired output. For data normalization, all data given to the deep autoencoder were presented as a response function in percentage units by dividing them into integral values of themselves and multiplying them by 100. In general, noise signals are added to the dataset with additional data processing procedure for an autoencoder to have the ability of noise reduction. In our problem, fluctuations in spectral data are coming from not noise signals but statistical uncertainties. By generating dataset via random sampling with randomly selected number of samplings, we can generate dataset with various level of counting statistics without additional procedure.

To compare reconstruction results with desired spectral data, mean absolute percentage error (MAPE) was used as a loss function, as described by Equation (3),
(3)$$\mathrm{MAPE}=\frac{100\%}{n}\sum_{i=1}^{n}\frac{O_{i}-I_{i}}{I_{i}}$$
where *n* is the number of channel bins, *i* indicates the *i*th channel bin, *O* is the Compton edge reconstructed spectrum, and *I* is the ideal spectrum given as desired output. 

MAPE was employed for the following reason. Although there are various options for the loss function, most of them represent difference rather than relative difference between two data sets. Because the data used in this study are plastic gamma spectra, they have relatively high levels of counts in low and high channels. Therefore, other options are mostly affected by values in the low channel region, and values in the high channel region tend to be ignored. However, MAPE represents the relative difference between two data because the subtraction of two data is divided by one of them. Therefore, it can calculate the difference between two data with equivalent weights for the whole region of spectral data whether the level of count is high or low.

The deep autoencoder was trained with the ADADELTA optimizer [27] for established training and validation sets during 1000 epochs. Model checkpoint option was activated as a callback function to save the best model built during the training procedure by monitoring validation loss, and the best model in the training procedure was used as the final model. Figure 5 shows a schematic illustration of the training procedure of our model, and Figure 6 illustrates the training and validation losses during the training procedure.

The performance of Compton edge reconstruction for the trained deep autoencoder was tested using the generated test set. Averaged test loss was 20.019 for test sets. Figure 7 shows Compton edge reconstruction results for several spectra of single and multiple radioisotopes. The deep autoencoder reconstructed the Compton edges in plastic gamma spectra, even though the spectra contains statistical uncertainties. Information on spectra and their corresponding MAPE values are presented in Table 1. Synthesis ratios in Table 1 were not estimated by the deep autoencoder, but rather acquired during the test set generation procedure.

### 3.2. Results of Compton Edge Reconstruction for Experimental Data

Reconstructions of Compton edges using the trained deep autoencoder were also conducted for the experimental data. In the environment described in Section 2.2, plastic gamma spectra were measured from single to multiple radioisotopes with a measurement period of 3600 s. Background radiation was also measured, and background-subtracted measured spectra were provided as input data to our autoencoder. Figure 8 shows the results of Compton edge reconstruction for measured spectra of single and multiple radioisotopes. Compton edges marked in Figure 8 represent theoretical energies of each source calculated by the following equation [28] (p. 51),
(4)$$E_{CE}=E\left(1-\frac{1}{1+\frac{2E}{m_{e}c^{2}}}\right)$$
where *E* is the energy of incident photon and *m_e_c*^2^ is the rest-mass energy of the electron (511 keV).

As shown in Figure 8, the energies of Compton edges in the reconstructed spectra were matched with their theoretical values calculated by Equation (4).

### 3.3. Minimum Reconstructible Counts

Similar to minimum detectable activity [29], the number of counts required to reconstruct Compton edges in plastic gamma spectra should be verified. In previous studies on gamma (or pseudo gamma) spectroscopy, similar concepts were defined to evaluate performance according to the activity of radioactive sources or the number of counts in their detection systems [12,30]. However, these cannot be used directly in our study because of the differences in their detailed concepts. Instead, averaged MAPE was used as a quality factor to evaluate the minimum reconstructible count (MRC) of the trained autoencoder. For each radioisotope, averaged MAPEs between reconstruction results and reconstruction references were calculated as follows. First, measured spectra in Section 3.2 (i.e., input spectra in Figure 8a–c) were normalized and utilized as PDFs for generating test sets for MRC evaluation. Second, 100 spectra were generated as test sets with the procedure detailed in Section 2.2 for each number of counts. Third, Compton edges were reconstructed for the test sets. Fourth, MAPEs between reconstruction results and reconstruction references were calculated for 100 generated spectra, and the averaged MAPE value was calculated. In this study, the reconstruction results presented in Section 3.2 (i.e., reconstructed spectra in Figure 8a–c) were used as a reconstruction reference. The threshold for MRC was determined as 10% of the averaged MAPE by referring to Table 1. Whole steps for MRC evaluation above were iterated with increment of the number of counts with interval of 50 for each radioisotope. Figure 9 shows the averaged MAPE according to the number of counts for single-isotope gamma spectra. MRCs were determined as the counts of which averaged MAPEs were decreased to less than 10%. Table 2 shows the MRCs of the single isotopes, and Figure 10 shows examples of generated spectra and reconstruction results corresponding to each MRC. In this table, MRCs are higher in order of ^60^Co < ^137^Cs < ^22^Na < ^133^Ba. The reason why MRCs are different depending on radioisotopes may be related to the intensities of energies of emitted photons and combinations of radioisotopes. ^60^Co emits two energies of gamma rays with almost analogous ratios. However, ^22^Na emits two energies of photons at different ratios; the intensity for a photon of 511 keV is almost double that for a photon of 1275.4 keV. This means that a higher number of counts is required to extract features for Compton edge reconstruction on the Compton continuum generated by a photon of 1275.4 keV. In the same manner, ^133^Ba requires the highest number of counts for Compton edge reconstruction due to the complex Compton edges in the low-energy region. In the case of ^137^Cs, the MRC was higher than the MRCs of ^60^Co, even though it emits one energy of gamma rays. It may because higher number of counts are required to discriminate following cases; one is ^137^Cs and the other is small ratio of ^133^Ba and ^137^Cs. 

To validate the MRC evaluation results, we measured the background and each isotope for 10, 20, 40, and 80 s corresponding to MRCs of ^60^Co, ^137^Cs, ^133^Ba, and ^22^Na, respectively, and Compton edges were reconstructed from measured net spectra (i.e., background-subtracted spectra). Total net counts for each measured net spectra were not exactly the same as the MRCs but were within statistical uncertainties. Figure 11 shows the results of Compton edge reconstruction with experimental spectra for validating each MRC. Identical to Figure 8, Compton edges marked in Figure 11 were calculated by Equation (4).

## 4. Discussion

A deep autoencoder model was presented to reconstruct Compton edges in plastic gamma spectra. Our model was trained to reconstruct Compton edges in plastic gamma spectra, even though the spectra have poor counting statistics, by designing a dataset generation procedure. As shown by the experimental results, it successfully reconstructed Compton edges in plastic gamma spectra with statistical uncertainties. Therefore, it was possible to conduct direct pseudo gamma spectroscopy using Compton edge reconstruction results. Furthermore, the MRCs of single isotopes were evaluated with the metric of MAPE as a loss function of our model. 

Although our model shows good performance on Compton edge reconstruction in plastic gamma spectra, there are three limitations we are aware of: First, the autoencoder generates data-specific results, i.e., it generates wrong results for spectra on radioisotopes that are not included in the training set; in fact, this is a characteristic of machine learning methods. For example, if untrained radioisotope is given, the autoencoder generates a spectrum which is one of the trained radioisotope or mixture of trained isotopes. Second, MRCs may be increased according to the increase in types of radioisotopes. For example, we evaluated the MRC of ^60^Co as 650, the minimum value among three isotopes. If, however, a radioisotope emitting gamma rays of energies similar to those of ^60^Co with almost analogous ratios was included in dataset, the MRC of ^60^Co may be increased because more counts are required to distinguish ^60^Co from the isotope. Furthermore, the spectra we used as input are for bare source. In practice, distortion of spectra may occur because of the presence of material surrounding the source, and it may affect Compton edge reconstruction performance. Concerning these limitations, further study is necessary.

## 5. Conclusions

This paper proposed a neural network model to reconstruct Compton edges in plastic gamma spectra. Datasets for training and validation of our model were generated by Monte Carlo simulations, data synthesis methods, and random sampling techniques. Although our model was trained by only simulation data, it successfully reconstructed Compton edges in simulated and measured gamma spectra, even though the spectra has poor counting statistics. Concerning the performance of Compton edge reconstruction according to counting statistics, MRCs were evaluated, and it was found that MRCs were related to the complexity of energies and intensities for emitted photons. 

Many researchers have been reported methods for pseudo gamma spectroscopy such as energy windowing, F-score analysis, energy weighted, and inverse matrix algorithms. These researches excluding inverse matrix algorithm were able to find existence of radioactive materials from the patterns after spectral data processing, rather than identifying the energy of gamma rays incident on the detector. Even though inverse matrix algorithm was able to identify the energy of gamma rays from unfolded gamma-ray spectra from plastic scintillators, it does not work for spectra with poor counting statistics. However, our method allows conducting direct pseudo spectroscopy with the analysis of reconstructed Compton edges even though the spectra have poor counting statistics.

## Figures and Tables

**Figure 1 sensors-20-02895-f001:**
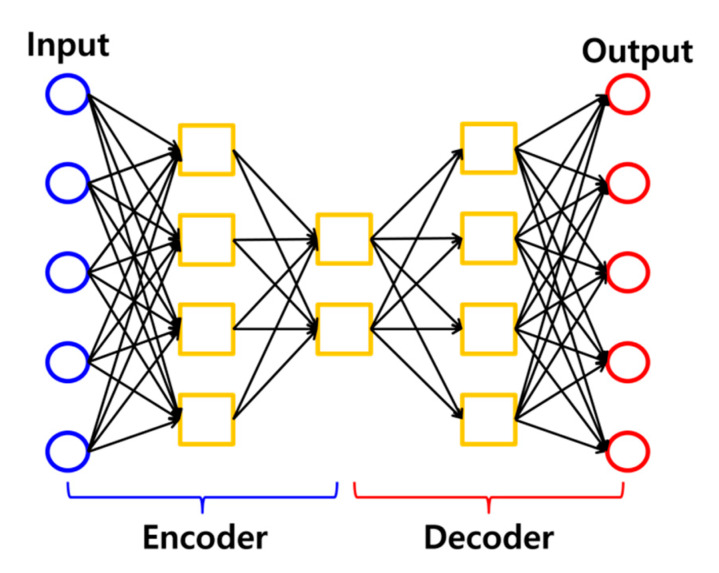
Schematic of autoencoder architecture.

**Figure 2 sensors-20-02895-f002:**
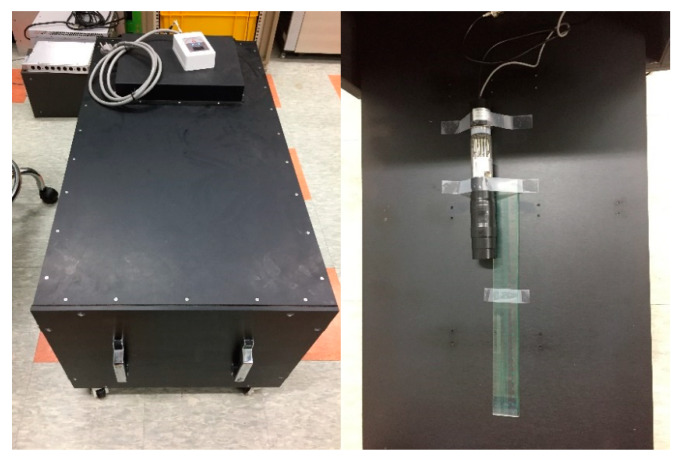
Aluminum dark box and experimental set-up.

**Figure 3 sensors-20-02895-f003:**
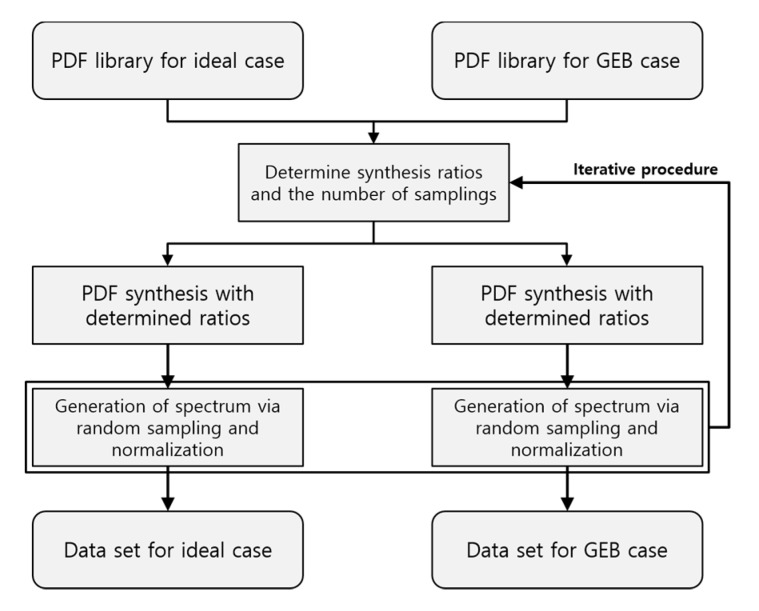
Flow chart of dataset generation.

**Figure 4 sensors-20-02895-f004:**
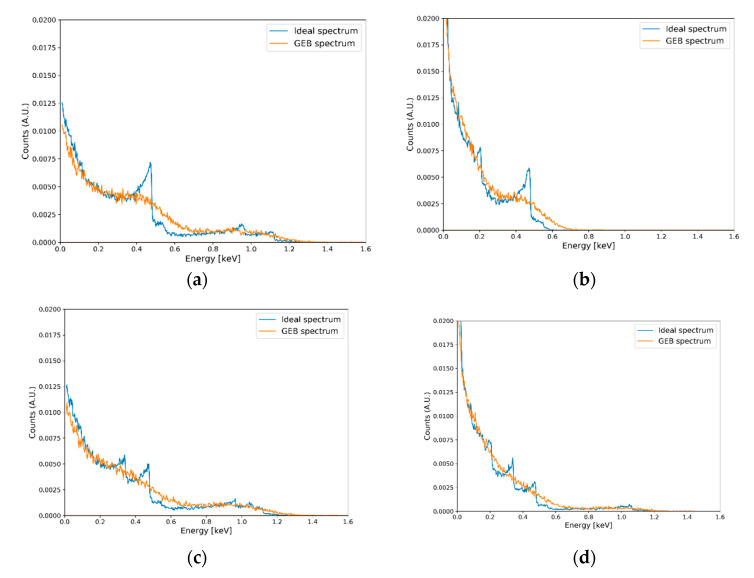
Examples of generated datasets for different synthesis ratios: (**a**) 70% ^22^Na and 30% ^60^Co; (**b**) 60% ^137^Cs, and 40% ^133^Ba; (**c**) 30% ^22^Na, 30% ^60^Co, and 40% ^137^Cs; and (**d**) 40% ^22^Na, 30% ^133^Ba, and 30% ^137^Cs.

**Figure 5 sensors-20-02895-f005:**
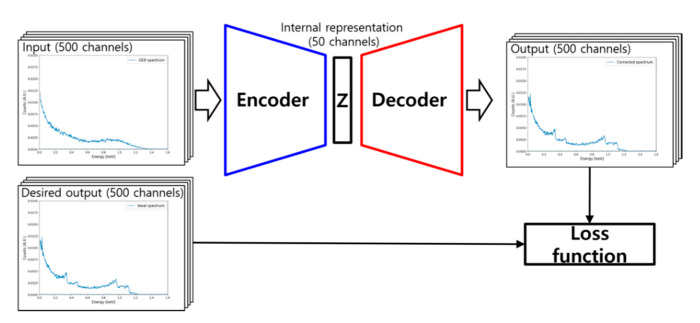
Training procedure of deep autoencoder.

**Figure 6 sensors-20-02895-f006:**
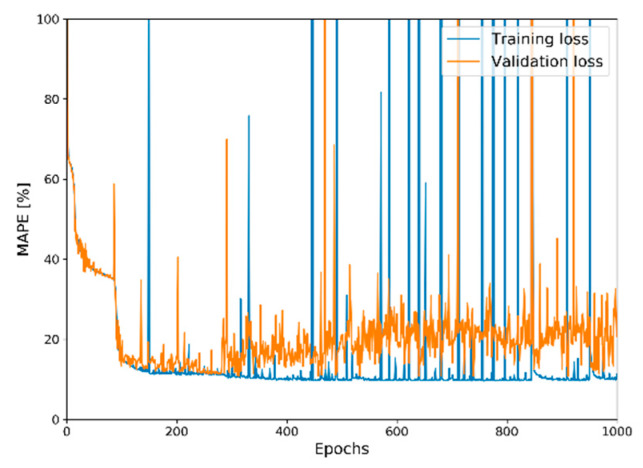
Historical plot of training and validation losses during training procedure. The best model during the epochs was stored with the use of the model checkpoint option and utilized as the final model.

**Figure 7 sensors-20-02895-f007:**
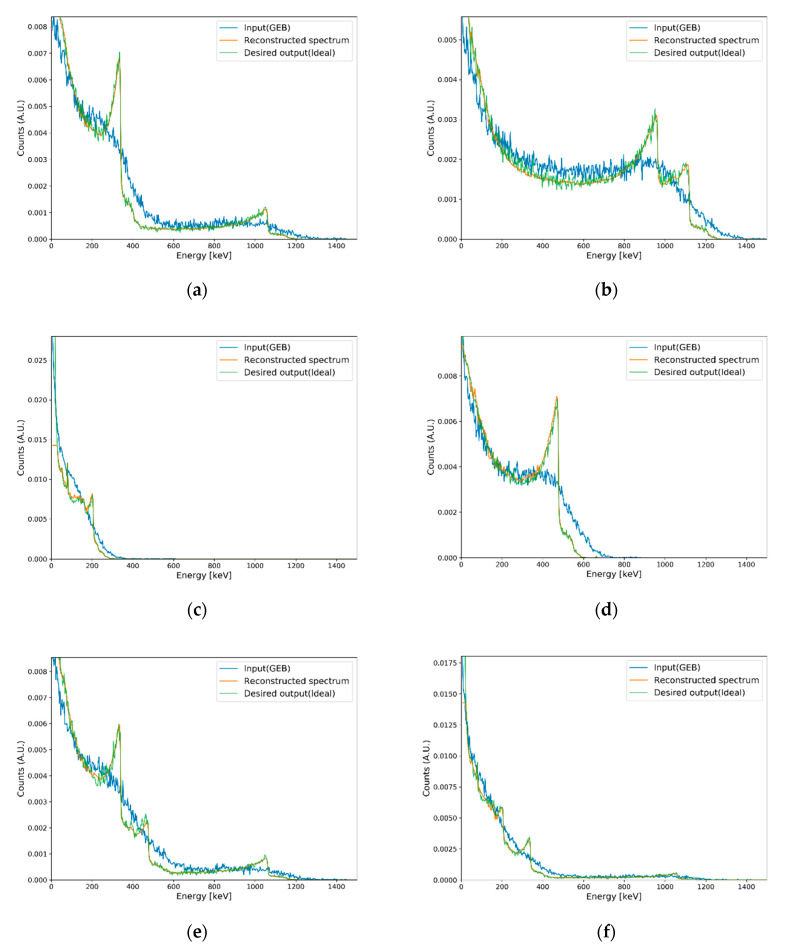
Results of Compton edge reconstruction for eight cases in the test set. Each synthesis ratio is (**a**) 100% ^22^Na; (**b**) 100% ^60^Co; (**c**) 100% ^133^Ba; (**d**) 100% ^137^Cs; (**e**) 70% ^22^Na and 30% ^137^Cs; (**f**) 50% ^22^Na and 50% ^133^Ba; (**g**) 20% ^60^Co, 20% ^133^Ba, and 60% ^137^Cs; and (**h**) 40% ^22^Na, 30% ^133^Ba, and 30% ^137^Cs.

**Figure 8 sensors-20-02895-f008:**
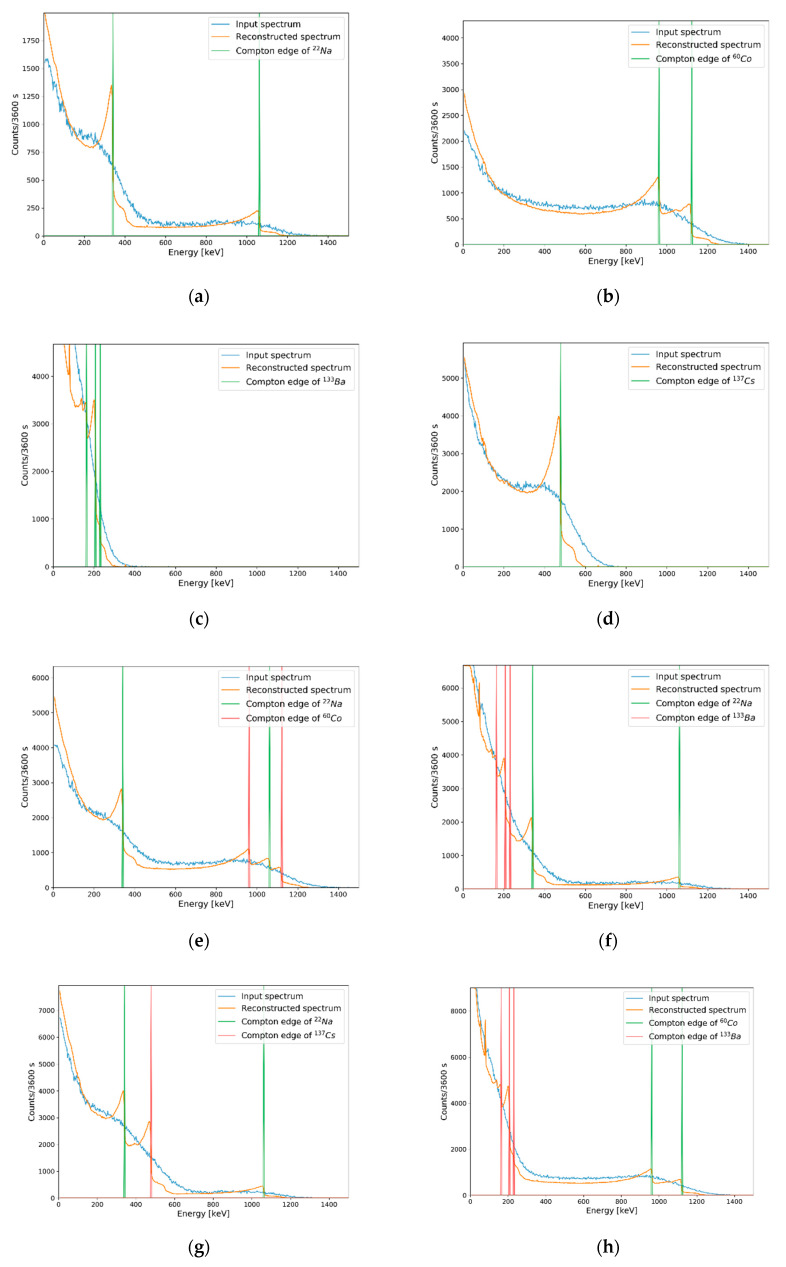
Results of Compton edge reconstruction for experimental data. Each figure represents measured spectra of (**a**) ^22^Na; (**b**) ^60^Co; (**c**) ^133^Ba; (**d**) ^137^Cs; (**e**) ^22^Na and ^60^Co; (**f**) ^22^Na and ^133^Ba; (**g**) ^22^Na and ^137^Cs; (**h**) ^60^Co and ^133^Ba; (**i**) ^60^Co and ^137^Cs; (**j**) ^133^Ba and ^137^Cs; (**k**) ^22^Na, ^60^Co and ^133^Ba; (**l**) ^22^Na, ^60^Co and ^137^Cs; (**m**) ^22^Na, ^133^Ba and ^137^Cs; (**n**) ^60^Co, ^133^Ba and ^137^Cs; and (**o**) ^22^Na, ^60^Co, ^133^Ba, and ^137^Cs.

**Figure 9 sensors-20-02895-f009:**
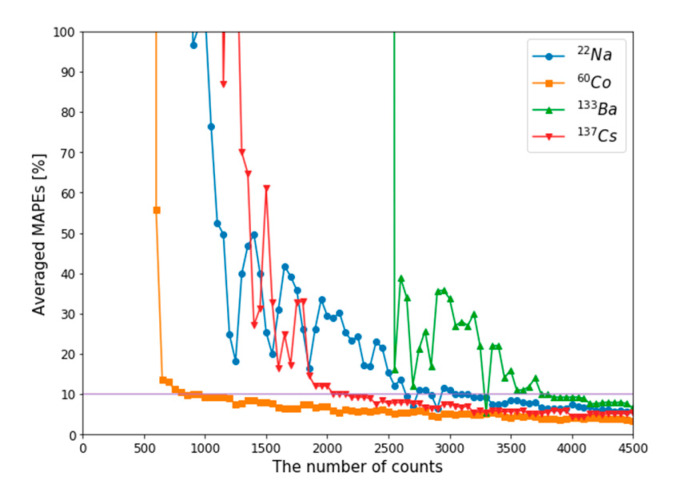
Averaged MAPEs according to the number of counts for single isotope gamma spectra.

**Figure 10 sensors-20-02895-f010:**
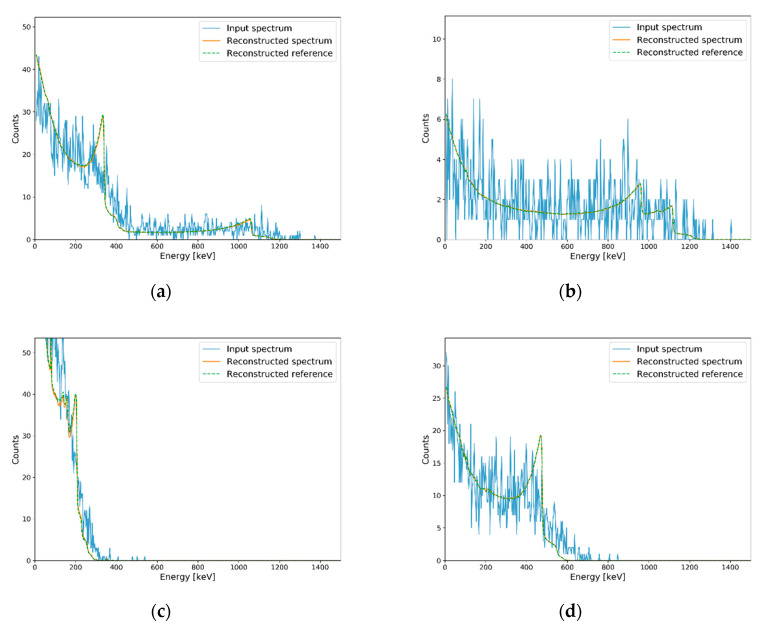
Examples of generated spectra for single isotope corresponding to each MRC and their Compton edge reconstruction results. Reconstruction results on (**a**) generated spectrum for ^22^Na, (**b**) generated spectrum for ^60^Co, (**c**) generated spectrum for ^133^Ba, and (**d**) generated spectrum for ^137^Cs.

**Figure 11 sensors-20-02895-f011:**
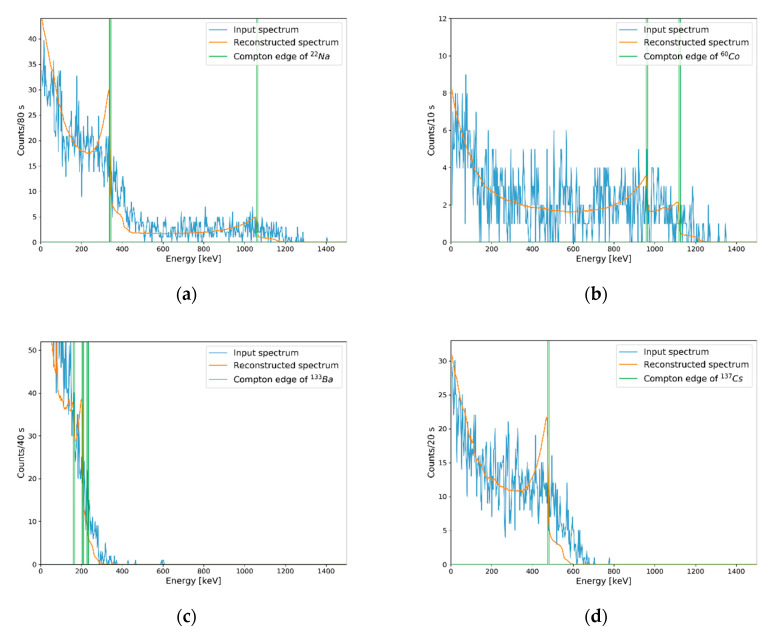
Results of Compton edge reconstruction for experimental spectra with different measuring periods corresponding to each MRC. Reconstruction results on (**a**) measured spectrum for ^22^Na during 80 s, (**b**) measured spectrum for ^60^Co during 10 s, (**c**) measured spectrum for ^133^Ba during 40 s, and (**d**) measured spectrum for ^133^Cs during 20 s.

**Table 1 sensors-20-02895-t001:** Information on seven cases in test set and their corresponding mean absolute percentage error (MAPE) values.

Case	The Number of Samplings	Synthesis Ratio				MAPE [%]
		γ_Na_	γ_Co_	γ_Ba_	γ_Cs_	
a	76,310	1.0	0.0	0.0	0.0	5.499
b	78,240	0.0	1.0	0.0	0.0	5.025
c	58,955	0.0	0.0	1.0	0.0	10.534
d	56,272	0.0	0.0	0.0	1.0	3.138
e	81,944	0.7	0.0	0.0	0.3	12.374
f	61,253	0.5	0.0	0.5	0.0	8.363
g	59,065	0.0	0.2	0.2	0.6	8.438
h	83,065	0.4	0.0	0.3	0.3	9.716

**Table 2 sensors-20-02895-t002:** Determined minimum reconstructible counts (MRCs) where averaged MAPEs are lower than 10% for each isotope.

Radioisotope	Energy [keV] [31]	Emission Intensity [%] [31]	MRC [#]
^22^Na	511	179.8	3050 ± 55
	1274.5	99.9	
^60^Co	1173.2	99.9	650 ± 25
	1332.5	99.98	
^133^Ba	53.16	2.14	3750 ± 61
	79.61	2.65	
	80.99	32.9	
	276.4	7.16	
	302.9	18.34	
	356	62.05	
	383.8	8.94	
^137^Cs	661.66	85.21	2000 ± 44

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
