# Peer review of "Reconstruction of Compton Edges in Plastic Gamma Spectra Using Deep Autoencoder"

_sensors, 2020, doi:10.3390/s20102895_

Round 1
Reviewer 1 Report
General comments:
Dear Authors,
This paper proposes a method to identify Compton edges in gamma-ray light output spectra as an attempt to obtain spectroscopy information from organic scintillators, which generally yield relatively smooth spectra, because of their detection mechanism, which is based on Compton scattering. The computational method of encoder-decoder network is well established and seems to be appropriate to this application.
One aspect that needs to be addressed in further detail is the sensitivity of the method to the detector resolution, which changes with the detector size and shape, other than material, or potential non-controlled bias in the gain, which would result in an energy shift in the spectrum. Additionally, the studied scenarios are not particularly challenging, could the authors add an analysis on nuclides such as Eu-135 or actinides?
Overall, the paper is well‑written, clear, and provides enough details on the methods. I recommend to add a summary of the quantitative results in the abstract.
Methods to perform “pseudo spectroscopy” with organic scintillators have been developed and have been commercially available for several years. I recommend that the authors mention these methods, referring for example to datasheets of commercial products or reports, e.g., https://www.pnnl.gov/main/publications/external/technical_reports/PNNL-18181.pdf , and refrain from stating “Differently from previous researches, it was possible to conduct direct pseudo spectroscopy with the analysis of Compton edges reconstructed by our method.”
Please make sure that the paper does not contain typographical errors, such as 60CO, instead of 60Co, or “Compon” instead of “Compton”. When appropriate, please refer to gamma-ray detection, instead of gamma detection.
Specific comments
The abstract would benefit from a summary of the quantitative results.
Line 69-70 please add the reference to the PMT data sheets
Line 71 please add a picture of the experimental setup
Line 87 please add details on the F8 tally
Line 88 Please avoid the use of the passive voice, for example please rephrase the sentence “F8 tallies were defined with and without a Gaussian energy broadening (GEB) card, respectively” as “F8 tallies were defined with and without a Gaussian energy broadening (GEB) card to acquire ideal and energy-broadened pulse height distributions, respectively."
Figure 3 please increase the font size
Reviewer 2 Report
I like the manuscript, it is relatively short and to the point. However there are parts of the manuscript that I do not fully grasp, while in other places I have specific and detailed comments. I list them below:
Page 1, line 31: Could you please explain what pseudo spectroscopy is?
Page 3, line 82: A missing "a" should be inserted "...we implemented a simulation geometry "
Page 3, lines 87-90: It is not clear how big the Gaussian broadening is, or why you selected the value you selected. Also the equation in which a, b and c are needed should be given.
Page 3, line 98: You write that you use dependent random sampling. Do you always use the same order in which you determine the synthesis ratio? If yes, shouldn't that influence the "randomness"?
Page 3, line 98: You write that by randomly selecting the sampling numbers, dataset with various noise levels could be generated. That is not correct, however, using various sampling numbers you can assess the impace of counting statistics.
Page 3, line 112: You have used a training set, a validation set and a test set. What did you use the validation set for? Did you optimze hyperparameters - if so which ones and how? What impace did the optimization process have?
Page 4, figure 3: In connection to showing these images, it would be interesting to already know the theoretically calculated position of the Compton edge. Also, what are the small "peaks" at just above 1 MeV in a) and around 1 MeV in c)?
Page 4 line 127: What was the reason for selecting 200, 100 and 50 neutrons in the encoder layer and 100, 200 and 500 neurons in the decoder layer? Did you try other values?
Page 4, line 131-132: What is a ReLU function and what does it do? Why was a sigmoid function used for the third layer in the decoder? Why did you choose these functions?
Page 4, line 136: "In general, noise signals are added ..." What do you mean by "in general" and what type of noise do you refer to?
Page 4, line 143: You write that "...I is the ideal spectrum given as desired output". I assume that you mean it is the actual value you are trying to predict, while "O" is the reconstructed spectrum? You write that "...O is the Compton edge reconstructed output", which could be (mis)interpreted as position of the Compton edge and not a full spectrum. Also, should you not calculate (I-O)/I rather than (O-I)/O, if I is the actual value and O is your prediction?
Page 4, line 148: "majorly"->"mostly"
Page 4, line 150: "into"->"by"
Page 4, line 154-156: I am not familiar with the ADADELTA optimizer, nor with "Model check point option". Is the latter something predefined? What does it do and how? Also you write that "...to save the best model...". How is the "best" model defined? When you write that "...the best model in the training procedure ws used as he final model", do you mean for the test data? Still there is no mentioning in the text of how the validation dataset was used.
Page 5, figure 5: This is the first time I see something about the validation set. But I still don't understand really where the figure comes from or what it is used for. What was the use of the validation dataset, which parametres were changed and how? How should figure 5 be interpreted?
Page 5, line 166: You write that the deep autoencoder reconstructed the Compton edges and improved the counting statistics, yet there are no results on counting statistics in the manuscript. Can you insert some to substantiate your statement?
Page 6, table 1: I am not sure I understand the table. The caption says that the table contains information on the seven cases in the test set, and the table shows roughly 45000-55000 samplings per case. In the beginning of the manuscript, you wrote that there were 2000 test spectra. So how many test spectra are there per case (about 2000/7~280?)? Is each spectrum then sampled about 175 times?
Page 7, line 188: remove "of themselves"
Page 7-8, figure 7: The legend is the same for all subplots, although there is many cases are no sources present of the type mentioned in the legend. There should be no legend associated with a source that is not present in the experiment.
Page 8, line 215: I assume that the reconstructed references are the actual values while the reconstructed results are the predictions? You also write that the average MAPE values was calculated. Average of what?
Page 8, line 216: "MAPEs between reconstruction results and reconstruction references were calculated, and the averaged MAPE value ..." Do you mean the average MAPE for each radioisotope? Please be more explicit about when you twrite that you calculate an average (average of what).
Page 8, line 219:Why did you determine a threshold at 10% of the averaged MAPE value?
Page 8, line 230: The weak peaks below 40 keV for Cs137, what are they?
Page 8, line 230: Here it is mentioned that weak peaks below 40 keV make it difficult to reconstruct the Compton edge. Can you explain what how you treat the X-rays?
Page 9, table 2: Where are the emission intensities taken from? There is no reference given. Also, some of these emission intensities (e.g. 4.46 and 36.36 do not agree very well with decay radiation provided by NNDC at BNL, see for instance their interactive nuclide chart). Also, there are some additional X-rays (see e.g. Co60) at low energies which are not included in the table, why?
Page 9, figure 8: The bin width makes it very difficult to see the green "line". Can you perhaps change the bin width or show an average over some bins to make this figure easier to "see"? Also, "which counts" in the spectrum are the ones on the x-axis in this plot? Did you limit the reconstructed counts in each spectrum to 100, 101, 102...5000 counts and calculate the MAPE, or did you exclude some counts (which?) from a "full" spectrum with 5000 counts in order to calculate the MAPE for lower number of counts? Maybe you could also add information on the total number of counts in some of the spectra in the paper so that it is easier to understand how smooth or rugged the spectra is for a certain number of counts?
Page 10-11, figure 10: The legend is the same for all subplots, although there is many cases are no sources present of the type mentioned in the legend. There should be no legend associated with a source that is not present in the experiment.
Page 11, line 273-280: I think this is very important input, but there is nothing more on it in the paper - there are no results on it earlier in the paper and no outlook that mentions what will be done about it in the future. Is it sufficient for your application to only be able to detect these three sources, or what do you plan to do about it?
Page 12, line 289: What do you mean with "the complexity of intensities of energies"?
Page 12, line 290: "researches" -> "researchers"?
Page 12, line 297: I noticed in the "Funding" section that this research is funded by the ministry of oceans and fisheries, but I have no idea how the research connects to the funder. Can you explain this? If you could perhaps describe that in the befinning of the paper it would be even better. I did not find anything in the introduction about specific applications or fields in which this research is particularly interesting.
Page 13, line 362: Probably a "method" too much.
Reviewer 3 Report
The article titled “Reconstruction of Compton Edges in Plastic Gamma Spectra using Deep Autoencoder” presents a method to reconstruct Compton edges using a machine learning –artificial neural network – method. The developed algorithm is used to locate Compton edges and identify radioactive sources. The ANN algorithm was trained and validated using Monte Carlo simulated spectra. The performance was verified using measured data.
Broad comments:
1/The introduction and abstract mention the challenging aspects and issues with plastic (or more generally organic) scintillators. While those are correct, the authors should also add a few sentences with the reason why those scintillators are used (low price, robustness, no cooling required, can be made in large sizes, fast, dual neutron-gamma response, …). This is dependent on applications and maybe the authors could give some context by tying it to a specific example.
2/The authors rightly acknowledge a major limitation of the present method: the algorithm is specific to the particular radioisotopes considered. The algorithm won't recognize an unknown Compton edge. This brings several issues that are not discussed in the manuscript:
a/ What happens if a different source is present. Will the algorithm wrongly attribute the presence of one of the three sources simulated?
b/ How does this algorithm behave when more sources are incorporated? Finding some applications where the number of sources is limited and demonstrating that the algorithm works under those conditions would bring a lot of value.
3/ Several other methods (based on machine learning or not) are mentioned in the introduction. It would be valuable to compare the MAPE value obtained with some of those other methods when applicable. For example, does the present algorithm provide any improvement from algorithm like the one described in Ingersoll and Wehring: https://www.sciencedirect.com/science/article/abs/pii/0029554X77904013
The article by Hamel et al. is probably also of interest though they are looking at a plastic loaded with Pb: https://ieeexplore.ieee.org/abstract/document/6727889
4/The sources spectra used as input are for bare sources. For sources like Na-22 that are not mono-energetic, the relative intensity of the gamma rays and therefore the relative number of counts from each gamma-ray energy will be dependent on the presence of material in front of the source. Similarly, scatter from surrounding materials will distort the spectra. This brings many problem for application-type scenarii.
Specific comments:
5/ l.31: This sentence should be changed to mention that the absence of full energy peaks in organic scintillators is in the region of interest (100 keV and above?). A photopeak is visible for lower energies like the for 59.5 keV line from Am-241 (still with poor resolution but a photopeak is present).
6/ l.51-52: I don't understand how the statistical uncertainties are reduced
7/ l63-64: It is unclear if it is three layers for the decoder and three layers for the encoder or three layers total
8/l.69-70: please provide which model from Eljen Technology was used. All of their scintillators have a PVT base, not PS.
9/l.73: please provide the time constant of the shaping amp
10/ l.90: please provide the a,b,c parameters obtained
11/l.104-105: The term “noise level” should be changed with something more descriptive. Really the only change is the number of events selected and therefore a pure statistics term. Noise levels could be added by the addition of a random background or other signals that are not due to the radioactive source.
12/ Figure 3. and general comment regarding the reconstruction of measured data and the calibration of the gamma-ray spectra in Ref. 19: Is it reasonable to look at the full energy range of the spectra? The very low energy part is clearly not reproduced well and could result in biases in the determination of the Compton edge. This is a known issue in the calibration of organic scintillators and most authors do generally constrain the fit of their data vs simulation around the Compton edge.
13/ Table 2: Is it reasonable to include the X-ray from Cs-137? Those lines will be on top of a predominant Compton edge. If the authors want to include those lines, particular care needs to be taken about electron light non linearity at those energies. This article by Moses et al. measured the electron light non-linearity for EJ-200: https://ieeexplore.ieee.org/abstract/document/6164292
Round 2
Reviewer 1 Report
Thank you for undertaking the suggested modifications.
Author Response
Thanks for your kind and considerate comments.
We believe your comments contribute to improve quality of our paper.
Reviewer 3 Report
The authors answered all my questions and made appropriate changes to the manuscript. It reads well and I only have a few very minor extra comments/suggestions. I labeled (OP) suggestions that are optional. I do not think the article requires further review.
l.34-35: "Therefore it is hard to conduct radioisotope identification, which is one of the major spectroscopic applications, from original plastic gamma spectra". I don't think the striked out part is correct. Radioisotope identification and gamma spectroscopy cannot be considered a major spectroscopic application for organic scintillators. I can see how it becomes of major interest for radiation portal monitors. Also I am not sure I understand the meaning of "from original plastic gamma spectra".
Plastic and more generally organic scintillators are scintillators of choice for beta spectroscopy and fast neutron detection (Knoll, Radiation detection and measurement", Chapter 8, Introduction). Their speed can also make them useful for high radiation environments. Brooks, "Development of organic scintillators" NIM 162 (1979) in part 4 gives a run down of the main applications of organic scintillators.
(OP) l52 & 58 and later on: "plastic gamma spectra" should be replaced by "gamma-ray spectra", or "gamma-ray spectra from plastic scintillators"
l59-60: "even though the spectra contain statistical uncertainties". What do you exactly mean here? Do you mean: "even from spectra with low counting statistics" ?
(OP) l77: "EJ-200 crystal" I don't think it is correct to refer to EJ-200 as a "crystal". The reader might also think of actual scintillation crystals like stilbene or anthracene.
(OP) l130: "even from spectra which has poor counting statistics" replace by "event from spectra with poor counting statistics"
(OP) l338 "plastic gamma spectra, it does not work spectra with poor counting statistics." should be replaced with " gamma-ray spectra from plastic scintillators, it does not work FOR spectra with poor counting statistics.
Author Response
Thanks for your kind and considerate comments. We believe your comments contribute to improve the quality of our paper.
Based on your comments, we revised our manuscript as you suggested.
l.34-35:
We revised the sentence as follows.
From
"Therefore, it is hard to conduct radioisotope identification from plastic gamma spectra. "
To
"Therefore, it is hard to conduct radioisotope identification, which is one of the major spectroscopic applications, from original plastic gamma spectra. "
(OP) l52 & 58 and later on: "plastic gamma spectra" should be replaced by "gamma-ray spectra", or "gamma-ray spectra from plastic scintillators"
Respond) We decide to keep the expression "plastic gamma spectra"
l59-60: "even though the spectra contain statistical uncertainties". What do you exactly mean here? Do you mean: "even from spectra with low counting statistics" ?
Respond) We revised our manuscript as follows.
From
"For generated and measured plastic gamma spectra, it is verified that our model can reconstruct Compton edges from spectral measurement, even though the spectra contains statistical uncertainties."
To
"For generated and measured plastic gamma spectra, it is verified that our model can reconstruct Compton edges from spectral measurement, even from spectra with low counting statistics."
(OP) l77: "EJ-200 crystal" I don't think it is correct to refer to EJ-200 as a "crystal". The reader might also think of actual scintillation crystals like stilbene or anthracene.
Respond) We revised the sentence as you suggested.
From
"EJ-200 crystal (cylindrical shape, dia. 30 50 mm, EJ technology) coupled with a PMT (R2228, HAMAMATSU) [20] and a preamp (E990-501, HAMAMATSU) [21] was used as a plastic scintillation detector"
To
"EJ-200 (cylindrical shape, dia. 30 50 mm, EJ technology) coupled with a PMT (R2228, HAMAMATSU) [20] and a preamp (E990-501, HAMAMATSU) [21] was used as a plastic scintillation detector"
(OP) l130: "even from spectra which has poor counting statistics" replace by "event from spectra with poor counting statistics"
Respond) We revised the sentence as you suggested.
From
"This means that it is possible to build an autoencoder model with the ability to reconstruct Compton edges even from spectra which has poor counting statistics with the generated datasets. "
To
"This means that it is possible to build an autoencoder model with the ability to reconstruct Compton edges even from spectra with poor counting statistics with the generated datasets. "
(OP) l338 "plastic gamma spectra, it does not work spectra with poor counting statistics." should be replaced with " gamma-ray spectra from plastic scintillators, it does not work FOR spectra with poor counting statistics.
Respond) We revised the sentence as you suggested.
From
"Even though inverse matrix algorithm was able to identify the energy of gamma rays from unfolded plastic gamma spectra, it does not work spectra with poor counting statistics"
To
"Even though inverse matrix algorithm was able to identify the energy of gamma rays from unfolded gamma-ray spectra from plastic scintillators, it does not work for spectra with poor counting statistics"